# Deltoid Muscle Tension Alterations Post Reverse Shoulder Arthroplasty: An Investigation Using Shear Wave Elastography

**DOI:** 10.3390/jcm12196184

**Published:** 2023-09-25

**Authors:** Annabel Fenwick, Thomas Reichel, Lars Eden, Jonas Schmalzl, Rainer Meffert, Piet Plumhoff, Fabian Gilbert

**Affiliations:** 1Department of Trauma, Hand, Plastic and Reconstructive Surgery, University Hospital of Wuerzburg, Oberduerrbacher Str. 6, 97080 Wuerzburg, Germany; 2Department of Trauma, Orthopedic, Hand and Plastic Surgery, University Hospital of Augsburg, Stenglinstrasse 2, 86156 Augsburg, Germany; 3Muskuloskelettales Universitätszentrum München, Unfallchirurgie, Ludwig-Maximilians-Universität München, Marchioninistr. 15, 81377 München, Germany; 4Department of Trauma, Shoulder and Reconstructive Surgery, Krankenhaus Rummelsberg GmbH, Rummelsberg 71, 90592 Schwarzenbruck, Germany; 5Ortho Höchberg, Zentrum für Orthopädie und Unfallchirurgie, Hauptstraße 78, 97204 Höchberg, Germany

**Keywords:** shear wave elastography, reverse shoulder arthroplasty

## Abstract

Introduction: This study aimed to evaluate the utility of shear wave elastography (SWE) in assessing changes in deltoid muscle properties following reverse shoulder arthroplasty (RSA). Methods: Our cohort consisted of 18 patients who underwent RSA due to various conditions, including osteoarthritis, cuff arthropathy, and irreducible proximal humeral fractures. Pre- and postoperative muscle elasticity and stiffness were measured using SWE and were compared with functional outcomes and radiological parameters. Results: Our results showed significant changes in deltoid muscle elasticity after RSA, particularly in the anterior and middle portions. However, these alterations were not correlated with postoperative functional outcomes or specific radiological parameters. The study also underscored the potential of SWE for future applications, including the preoperative assessment of deltoid function, postoperative monitoring, and intraoperative use for optimal component positioning during RSA. Conclusion: Further research, involving larger, more homogeneous patient cohorts is needed to confirm these findings and to explore the potential influence of these changes on the biomechanical design of implants and prosthesis positioning in RSA.

## 1. Introduction

The incidence of patients undergoing reverse shoulder arthroplasty (RSA) has been on a steady upward trend, fueled by the expanding list of indications such as cuff tear arthropathy and primary omarthritis, but also after failed arthroplasty and rheumatoid arthritis [1]. Patients presenting with shoulder pain are constantly increasing [2] and clinical approaches and pathways have been created to better differentiate between the underlying reasons for shoulder pain and help in the further management of suffering patients [3].

Due to the unique anatomical modifications in RSA, the deltoid muscle becomes critically integral to the postoperative functional outcome, irrespective of the underlying cause of the surgery—be it proximal humeral fracture, osteoarthritis, or rotator cuff arthropathy [2]. Several factors can potentially impede the postoperative outcome, including fatty muscle atrophy, acromion and glenoid fractures, malpositioning of the prosthesis, scapular notching, inappropriate tension on the deltoid muscle, and axillary nerve damage, all leading to prosthesis malfunction and patient dissatisfaction [4,5,6].

Ultrasound elastography, a popular method for tissue evaluation in liver, thyroid, or breast diseases, has gained interest in musculoskeletal applications [7,8,9]. Recent studies have argued its reliability in detecting changes in soft tissue properties due to varying conditions or pathologies [10,11]. Ultrasound elastography creates a new dimension to ultrasound imaging, which allows for a new evaluation of tissue quality [12]. It is based on the principle that the application of mechanical stress causes stiffness depending on changes in underlying tissue. Two primary types exist—strain and shear wave elastography. Shear wave elastography (SWEUS) was first described by Sarvazyan et al. in 1998 [13]. Shear wave elastography (SWE) has demonstrated lesser examiner dependence, easy accessibility, low cost, and fewer contraindications compared to MRI scans or contrast use [10,11,14,15,16,17]. The process calculates tissue stiffness after the application of an acoustic impulse, known as the Acoustic Radiation Force Impulse (ARFI), which deforms the underlying tissue. The resulting shear waves, the speed of which is proportional to tissue properties such as stiffness, are measured in m/s. This allows for the estimation of shear wave velocity and, through a mathematical equation, conversion to an elastic modulus. For reliable results, the probe must be aligned parallel to the muscle fibers, and no tissue compression is required during the exam. Thus, SWE can provide quantitative information on the elastic modulus of the examined tissues.

Recent investigations have particularly targeted the shoulder girdle, including the rotator cuff (especially the supraspinatus muscle), the deltoid muscle, and neck muscles [18,19,20,21]. As the deltoid muscle plays a critical role in the biomechanics of shoulders undergoing reverse shoulder arthroplasty (RSA), especially with regard to over-tensioning, both pre- and postoperative assessments of the deltoid muscle quality are very important but still remain challenging. EUS measurement of deltoid muscle elasticity revealed that operated-on deltoid muscles showed higher stiffness than contralateral healthy deltoid muscles [5]. However, only a few studies have incorporated clinical results or attempted to correlate these findings with the functional outcome. This study aims to evaluate the practicability of shear wave elastography in assessing the deltoid muscle following reverse shoulder arthroplasty, determine variations in deltoid stiffness compared to the contralateral side, and to understand its role as a determinant of functional outcomes.

## 2. Materials and Methods

Eighteen patients who had undergone reverse shoulder arthroplasty were enrolled in our study. In twelve of these cases, the arthroplasty was necessitated due to primary glenohumeral omarthritis (N = 3) or cuff arthropathy (N = 9), while the remaining six were undertaken due to dislocated, irreducible proximal humeral fractures. All surgeries were executed by three senior consultants specializing in shoulder surgery, using the deltopectoral approach.

The conducted study was approved by the local Ethics Committee and fulfilled the standards of the Declaration of Helsinki.

Patients were contacted via telephone and informed about the study, and written informed consent was obtained from each participant. Demographic data, such as gender, age, smoking status, Body Mass Index (BMI), and medical history, were retrieved from our clinical database.

Ultrasound and shear wave elastography examinations were conducted using Aixplorer (SuperSonic Imagine, Aix-en provence, France) with a 9 MHz linear array transducer, maintaining identical conditions for all procedures. A single physician performed these examinations on both shoulders of every patient in a sitting position with the arm resting palm downwards on the legs.

The three anatomical regions of the deltoid muscle (Pars clavicularis, pars acromialis, pars spinalis) were individually examined and measured with the ultrasound probe positioned parallel to the muscle fibers, as shown in Figure 1 [22]. Every region of interest was measured three times to calculate the average. The humeral neck was visualized as a reference for probe positioning, as shown in Figure 2. Each region of interest (ROI) in the deltoid on both sides was examined both in a resting position and under maximum abduction force. Instead of using multiple measurement points, the specific deltoid ROI was manually outlined without interference from fascias or bony substance, and the integral for the shear wave velocity and the elasticity modulus—as a reference for tissue stiffness—was calculated. The depth was set at 2 centimeters.

In addition, a force measurement was executed using IsoForceControl Ca. Medical Device Solutions AG, Oberburg, Switzerland [23] on the operated and contralateral shoulder, as well as the shoulder and elbow visual analog scale. The range of motion (elevation, abduction, internal and external rotation) was also measured for every patient.

For radiological evaluation, the most recent X-rays were examined. Anteroposterior (a.p.), axial, and Y views were obtained, and measurements for offset, retroversion, tilt, and acromio-humeral distance were calculated.

Statistical analysis was performed using SPSS version 18 (IBM, Armonk, NY, USA). Parameters were examined for normal distribution, and the level of significance for dependent samples was calculated using the Mann-Whitney U test and Kruskal-Wallis test. Pearson’s correlation was employed to assess the relationship between strength, clinical outcome, and tissue properties. Differences were considered statistically significant at a *p*-value less than 0.05.

## 3. Results

Our study consisted of thirteen female and five male patients, with an average age of 76 years (range, 64–84 years). In 66% of cases, the right side was operated upon. The dominant side was affected in 80%. Reverse total shoulder arthroplasty (RTSA) was performed using a prosthesis by Lima for proximal humerus fractures and Tornier/Wright, Aequalis/Ascend II for cuff arthropathy. The follow-up, inclusive of the ultrasound examination, was postoperatively conducted on average for 15 months (range, 4–48 months).

The cohort, with an average BMI of 25.9 (range, 22.1–30.1), leaned slightly towards obesity. Two patients were still active smokers. On average, each patient had at least two comorbidities, with over a third of the patients on multi-medication (>5 daily). In this small group, we were unable to correlate the comorbidities with the functional outcome.

The recorded functional outcome was generally satisfactory. All patients who underwent RSA due to osteoarthritis affirmed they would opt for surgery again if required. On average, patients scored 18 points on the Visual Analog Scale (VAS) for pain (range, 5–47) and 64 points on the VAS for function (range, 64–97). The Constant Murley Score averaged at 66 points (range, 35–89). Notably, patients who underwent RTSA due to cuff arthropathy demonstrated superior functional results compared to those suffering from proximal humeral fractures, and also reported a higher satisfaction rate. The average range of motion was 150° for abduction, 25° for internal rotation, 140° for elevation, and 25° for external rotation. Our study recorded no cases of infection or required revisions or periprosthetic fractures.

Force measurement was carried out on all patients. As anticipated, patients were unable to generate an equal amount of force on the RTSA side compared to the healthy side. On average, 48.8 N was achieved after RTSA, compared to 58.3 N on the healthy side as the maximum achievable strength. However, this difference was only marginally significant (*p* > 0.07). We did not establish any link between muscle elasticity and the postoperative functional outcome indicated by the scores.

A radiological assessment was performed on the latest available shoulder X-rays. On average, the prosthesis was positioned at an offset of 39.07 mm with a tilt of 3.2°. The glenosphere was size 36 in all but two cases, where it was size 42, and was implanted with a slight retroversion of 3°. The average acromio-humeral distance was 28.5 mm (range 18.2–33.9 mm).

Shear wave elastography (SWE) revealed a higher muscle tension in the deltoid in patients post RSA compared to the contralateral non-operated side across all examined areas under relaxed conditions. These differences were particularly noticeable in the pars clavicularis (PC) and pars acromialis (PA), with a statistically significant difference (*p* < 0.05), which was not apparent in the posterior part, pars spinalis (PS), as shown in Figure 3.

When measurements were repeated under isometric load, all regions of the deltoid muscle adapted to the new condition irrespective of RSA, with a significant increase in elasticity (*p* < 0.001), as shown in Figure 4. The increase in the elasticity modulus was higher after RSA than on the healthy side, as shown in Figure 5. However, no correlation was found between clinical and functional outcomes and the elastography findings.

## 4. Discussion

Recent studies in the literature have raised the question as to whether shear wave elastography (SWE) is a reliable method for detecting changes in muscle tissue properties, thereby serving as a method for measuring muscle activity and related pathologies. In a previous study, we demonstrated the value of SWE by showing a correlation with the MRI spectroscopic measurement of fatty degeneration, and we now intend to further explore the potential applications of SWE [24].

Kim et al. [25] evaluated twelve healthy participants during isometric movement of the shoulder girdle and concluded that SWE is an excellent method for evaluating muscle stiffness in both static and dynamic modes and demonstrates high inter- and intraobserver reliability. For this reason, we conducted our assessment of the deltoid after reverse shoulder arthroplasty (RSA) in both a resting position and under isometric loading, confirming the feasibility and usefulness of SWE.

Numerous studies have investigated muscle tissue properties and their pathologies around the shoulder joint and the neck, including the evaluation of normal stiffness values for the pericranial muscles, as well as changes in and values of the rotator cuff and in patients with adhesive capsulitis [14,25,26,27].

In our study, we focused on the properties of the deltoid muscle, particularly its changes after RSA. Hatta et al. [22] experimentally showed, on eight fresh-frozen cadaver shoulders, that endoscopic ultrasound (EUS) could be a reliable and feasible method to quantitatively assess the mechanical properties of the deltoid muscle by comparing elongated and native deltoid muscles. The results were especially promising in the anterior and middle portions. These results align with our findings that performing RSA results in the anterior and middle portion of the deltoid muscle showing a significant increase in shear modulus in contrast to the contralateral side.

Fischer et al. [28,29] examined the deltoid muscle of 64 patients treated with RSA using contrast-enhanced ultrasound (CEUS) as well as EUS and assessed muscle function using various shoulder scores. EUS measurement of the deltoid muscle elasticity revealed that operated deltoid muscles showed higher stiffness than contralateral healthy deltoid muscles, which also matches our results. They concluded from the CEUS results that perfusion was the primary impact on the postoperative results and deemed it a surrogate parameter.

Roche et al. [30] examined deltoid wrapping and tensioning of the various prosthesis designs and positioning in a computer model of the shoulder. On average, RSA resulted in an elongation of the deltoid of about 10 to 20%, siding with a functional shortening of external and internal rotators. Excessive tension can result in malfunction and a lack of motion, while enough tension is necessary for joint stability. We examined various radiological parameters deemed critical in the study (tilt, retroversion), but in this small cohort, we could not find any significant correlations with our findings in the ultrasound examination.

A further field of interest for SWE could be within the rehabilitation process after musculoskeletal disorders, such as the implantation of reverse shoulder arthroplasty. A study by Piccoli et al. showed the importance of the “Effect of Attentional Focus Instructions” on motor learning and performance, and SWE could be an important tool in the assessment of the deltoid and shoulder function during rehabilitation [31].

We believe that the value of SWE lies in potential future fields of use. It could serve as a tool for assessing the function of the deltoid muscle, including preoperative scanning and the selection of patients with good deltoid function who will benefit from RSA, and postoperative monitoring of deltoid function. Moreover, it could serve as a tool for analyzing complications in cases of malfunctioning prosthesis, potentially revealing reasons for acromion stress fractures or deltoid dysfunction.

Further applications could include the intraoperative use and assessment of deltoid elongation and tension, which could enable the optimal positioning of reverse arthroplasty components during surgery.

Whether our findings have a significant impact on postoperative functional outcome and, therefore, imply changes to biomechanical designs of implants or the positioning of the prosthesis will require further investigations and large randomized controlled studies.

The limitations of this study are its mono-centric design, small sample number, as well as the heterogenic cohort, and so conclusions have to be carefully drawn. Furthermore, the observed changes in muscle elasticity did not correlate with radiological parameters such as tilt and retroversion, nor with functional outcomes.

## 5. Conclusions

Shear wave elastography (SWE) has potential for its application as a reliable method for detecting changes in muscle tissue properties and assessing muscle activity. Our study provided evidence that the deltoid muscle undergoes significant changes in its elasticity after reverse shoulder arthroplasty (RSA), with increased stiffness noted when compared to the contralateral side. This increase was particularly evident in the anterior and middle portions of the deltoid muscle.

Further research is necessary to explore these possibilities and to ascertain whether our findings could influence the biomechanical design of implants or the positioning of the prosthesis in future RSA procedures. Large, randomized controlled studies will be crucial in this pursuit.

## Figures and Tables

**Figure 1 jcm-12-06184-f001:**
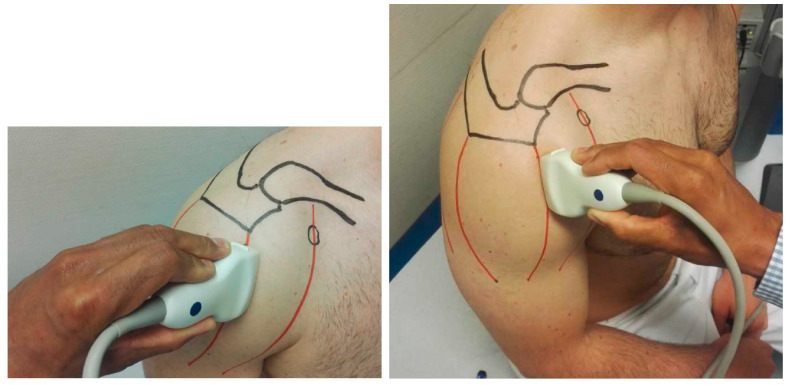
Standard examination set up.

**Figure 2 jcm-12-06184-f002:**
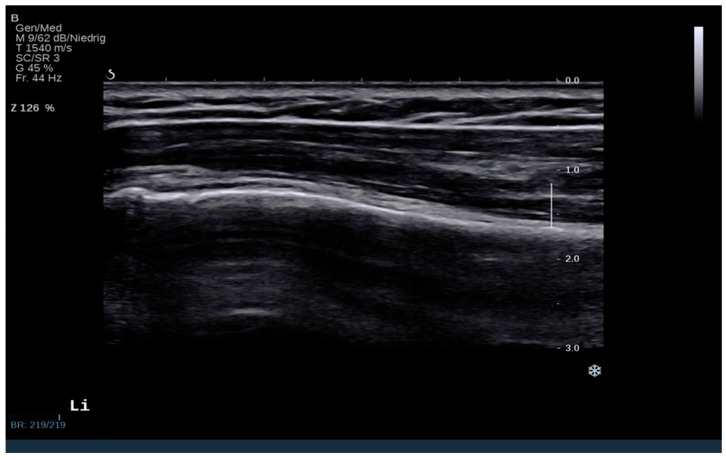
Example for standardized ultrasound B-mode slide including humeral neck and deltoid muscle, middle portion as standard examination setup.

**Figure 3 jcm-12-06184-f003:**
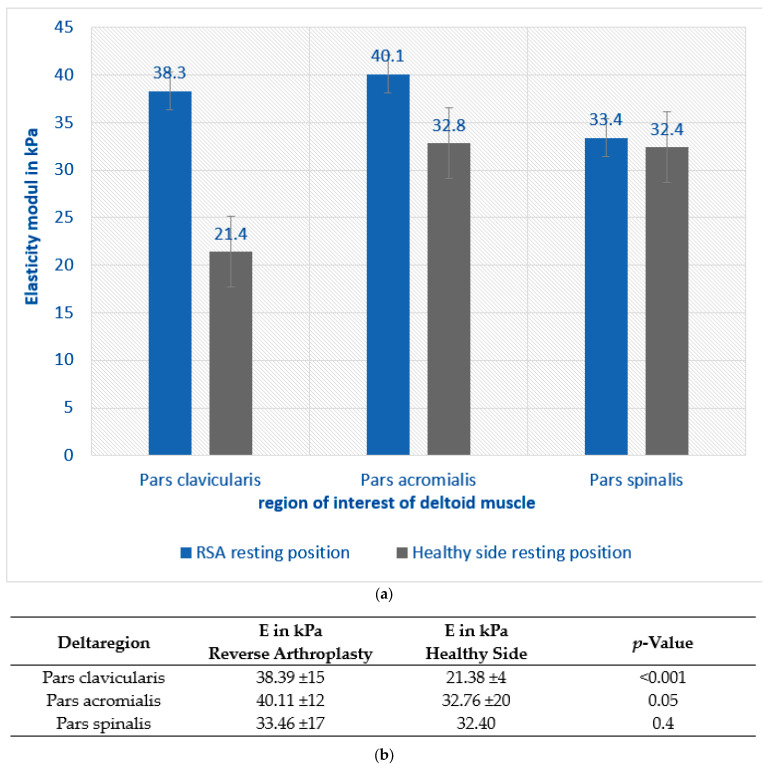
(**a**) Shear wave elastography shows a change in deltoid elasticity in each region of the deltoid with an increased tension after RSA than on the healthy contralateral side, especially in the anterior and middle portion of the deltoid in resting position without loading (*p* < 0.05). (**b**) Exact values for SWE in resting position.

**Figure 4 jcm-12-06184-f004:**
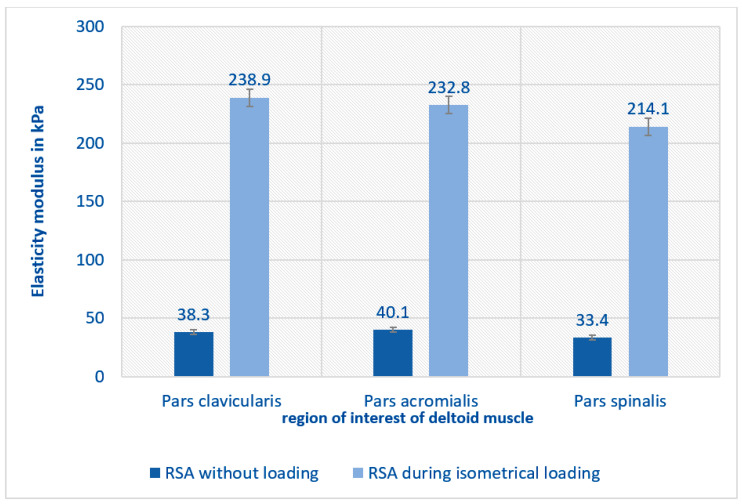
Isometric loading of the shoulder after RSA leads to a significant increase in tension of the whole deltoid muscle in comparison to elasticity measurements in a resting position in the same shoulder *p* < 0.005.

**Figure 5 jcm-12-06184-f005:**
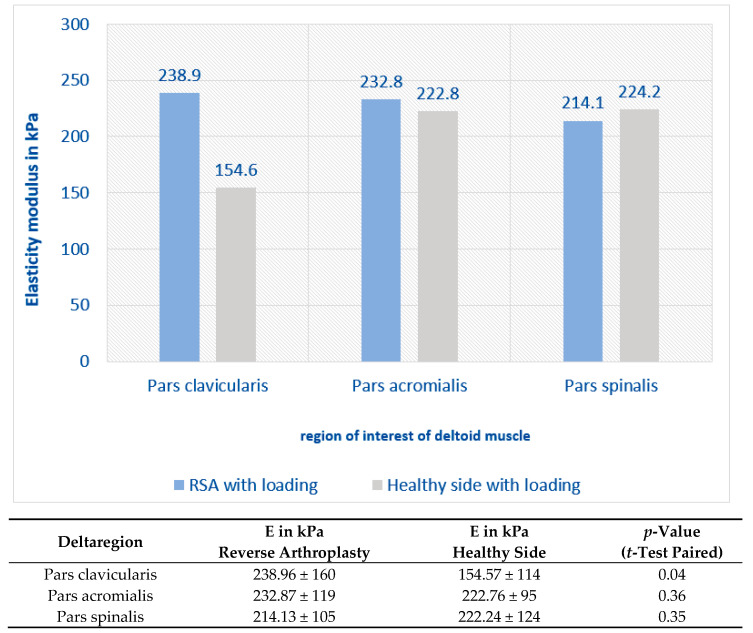
Isometric loading leads to a greater increase of the elasticity modulus after RSA than on the healthy contralateral side (only significant in the pars clavicularis).

## Data Availability

Ultrasound illustrations should be published in color.

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
