# Peer review of "Deltoid Muscle Tension Alterations Post Reverse Shoulder Arthroplasty: An Investigation Using Shear Wave Elastography"

_jcm, 2023, doi:10.3390/jcm12196184_

Round 1

Reviewer 1 Report

This paper aims to evaluate the utility of shear wave elastography (SWE) to assess the deltoid muscle elasticity following reverse shoulder arthroplasty (RSA), and aims to understand the role of the deltoid muscle in the outcome.

It is an interesting paper since the optimal soft tissue tension, and particularly the deltoid muscle, is unknown for RSA. Shear wave elastography seems to be a reproducible method to objectively assess the deltoid tensioning. However, the role of the deltoid muscle in functional outcome is not clearly answered in the paper. The cohort is very heterogen with different RSA indications and designs. Hence, it is a lot of variables to analyze specifically the role of the deltoid in the functional outcome.

This paper is interesting, but needs to be revise to answer with more clarity regarding the role of deltoid tensioning in functional outcome.

Material and Methods

- when was performed the SWE relatively to the RSA surgery?

- when you say “omarthritis” is it “primary gleno humeral omarthritis”?

- How many patient were with omarthritris? And CTA?

- This part would benefit from a figure or a picture to show the position of the probe and the arm

- do you have any reference for the measurement of force with flexibar?

- how do you measure the prosthesis offset and tilt?

Results

- what is the rate of dominant side?

- it would be great to have the functional outcomes for each etiology in a table

- did you find any difference in SWE results between each prosthesis design? Regarding arm lengthening

- was there any correlation between AHD and deltoid tension?

- what are the results of the SWE? a table with the results would be great

- is there any correlation between deltoid tensioning and the ROM? AFE / RE / RI?

- there is a comparison of results for CTA vs. proximal humeral fracture. What about PGHOA?

Author Response

see below word document

Reviewer 2 Report

Dear Authors

Thanks a lot for the opportunity you have offered me to revise the fascinating manuscript "Deltoid Muscle Tension Alterations Post-Reverse Shoulder Arthroplasty: An Investigation Using Shear Wave Elastography".

As a significant strength, this manuscript evaluates the practicability of shear wave elastography in assessing the deltoid muscle following reverse shoulder arthroplasty, determines variations in deltoid stiffness compared to the contralateral side, and understands its role as a determinant of functional outcome. This proposal is a novelty in the field and adds information to the existing evidence in the literature produced in the field.

As a major weakness, the manuscript sometimes needs few details and clarity concerning methodological steps that would help improve the understanding of the manuscript. 

Overall, the paper is well-structured, developed and written. Thus, my peer review is a minor revision. After integrating the improvements, I will be happy to reconsider it.

#INTRODUCTION

*Background: I suggest the authors report more details about the background of shoulder pain due to multifactorial causes leading to reverse shoulder arthroplasty. An example of a reference to be integrated into the background is the following. doi: 10.1186/s40945-018-0050-3.

#METHODS

*reporting: Reporting guidelines should be used for this observational study. E.g. STROBE (doi: 10.1371/journal.pmed.0040296). Please organise the reporting of methods accordingly.

*ethics: please, also report in the main text the details about ethics.

#RESULTS

*reporting: Reporting guidelines should be used for this observational study. E.g. STROBE (doi: 10.1371/journal.pmed.0040296). Please organise the reporting of results accordingly.

#DISCUSSION

*reporting: Reporting guidelines should be used for this observational study. E.g. STROBE (doi: 10.1371/journal.pmed.0040296). Please organise the reporting of the discussion accordingly.

*exercise: I suggest authors discuss the importance of studying the shear wave elastography in assessing the deltoid muscle following reverse shoulder arthroplasty after a functional rehabilitation that adopts a therapeutic exercise. An example of a reference to be integrated into the discussion is the following doi: 10.3390/jfmk3030040

*limitations: please, add a section about the limitations of this study (e.g., mono-centric, small sample…).

#CONCLUSION

*implications for practice and research: What are the practical implications for sports people and for future research? I suggest the authors include this part here in the discussions.

The english is good.

Author Response

see below document
